# GeValDi: Generative Validation of Discriminative Models

**Vivek Palaniappan**
University of Cambridge
vp392@cam.ac.uk

**Matthew Ashman**
University of Cambridge
mca39@cam.ac.uk

**Katherine M. Collins**
University of Cambridge
kmc61@cam.ac.uk

**Juyeon Heo**
University of Cambridge
jh2324@cam.ac.uk

**Adrian Weller**
University of Cambridge
aw665@cam.ac.uk

**Umang Bhatt**
University of Cambridge
usb20@cam.ac.uk

## Abstract

Evaluation of machine learning models is critically important for reliable use. Typically, validation datasets need to be large and are hard to procure, yet multiple models may perform equally well on such datasets. We offer GeValDi: a data-efficient method to validate discriminative classifiers by creating samples where such classifiers maximally differ. We demonstrate how such "maximally different samples" can be constructed and leveraged to probe the failure modes of classifiers.

## 1 Introduction

In a typical machine learning (ML) problem, several ML models may perform equally well on training datasets (Breiman, 2001), but substantially differently on unseen data. How then are we to choose which model to deploy? Distinguishing between similarly performing models requires large, validation datasets that are typically unavailable. We propose a data-efficient solution to probe the differences between comparably performing classifiers *without using validation data*, by *synthesising* data where they maximally differ. We call our approach GeValDi: Generative Validation of Discriminative classifiers. Using ImageNet (Deng et al., 2009) as a case study, we empirically investigate the ability of our approach to generate "maximally different samples."

## 2 GeValDi

In this section, we introduce GeValDi. Given a set of $M$ disriminative classifiers, defined by their predictive probabilities $\{p_m(\mathbf{y}|\mathbf{x})\}_{m=1}^M$, GeValDi generates data in the input space where the classifiers maximally differ, which is formalised as

$$\tilde{\mathbf{x}} = \arg\max_{\mathbf{x} \in \mathcal{X}} D\left[\{p_m(\mathbf{y}|\mathbf{x})\}_{m=1}^M\right] \tag{1}$$

where $\tilde{\mathbf{x}} \in \mathcal{X}$ denotes the *maximally different sample* (MDS), $\mathcal{X}$ denotes the input space and $D$ denotes some divergence measure, such as the Jensen-Shannon (Lin, 1991) or KL (Kullback & Leibler, 1951) divergence. However, without restricting the search space, optimising this objective will return data on which our classifiers are unlikely to be evaluated. Our key insight is to constrain the search by optimising in the latent space of a generative model, $p(\mathbf{x}) = \int p(\mathbf{x}|\mathbf{z})p(\mathbf{z})d\mathbf{z}$ with prior $p(\mathbf{z})$, and conditional distribution $p(\mathbf{x}|\mathbf{z})$ with mean function $E[p(\mathbf{x}|\mathbf{z})] = g(\mathbf{z})$. We can formalise the problem of generating an MDS with high-probability under the true data distribution as

$$g^{-1}(\tilde{\mathbf{x}}) = \arg\max_{\mathbf{z} \in \mathcal{Z}} D\left[\{p_m(\mathbf{y}|g(\mathbf{z}))\}_{m=1}^M\right] + \lambda \log p(\mathbf{z}) \tag{2}$$

where $\lambda$ trades off flexibility of the optimisation problem with the likelihood of the generated sample under $\tilde{p}(\mathbf{x})$. By optimising in the low-dimensional latent space, we ensure samples are from the data manifold. For the full algorithm, please see Appendix A

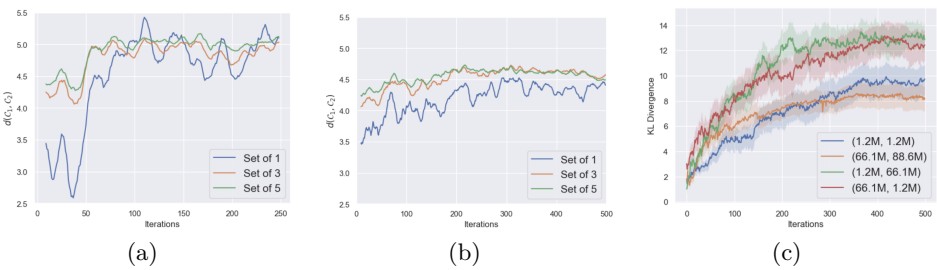

Figure 1: Each figure shows images generated before (left) and after (right) the latent space optimisation for Model 1 - GoogleNet (Szegedy et al., 2014) vs Model 2 - AlexNet (Krizhevsky et al., 2012). Bolded text highlights conditional class-labels in BigGAN.

Figure 2: (a) and (b) show evolution of set distance for top $n = 1, 3, 5$ predictions for a pair of models, where number of parameters (model 1, model 2) are (1.2 million, 1.2 million) and (66.1 million, 88.6 million) respectively. (c) shows evolution of KL divergence for four pairs of models of varying expressivities.

## 3 RESULTS

We study the ability of GeValDi to generate realistic synthetic samples using BigGAN (Brock et al., 2018) for which the predictions of two high-performing classifiers differ, on ImageNet (Deng et al., 2009). A complete list of experimental details can be found in Appendix C.

In Figure 1, we compare the optimised and pre-optimised MDS images, along with predicted class probabilities for our two classifiers. As expected, whilst photo-realistic, the optimised MDS deviate noticeably from the pre-optimised images. Furthermore, we are able to find failure modes of high-performance classifiers using synthetic data. For example, in Figure 1(a), although the final image is clearly a lifeboat, GoogleNet incorrectly classifies it as a drilling platform. In Figure 2 shows evolution of KL divergence, and prediction set distance for models with different expressivities. "Prediction set distance" measures the distance between two sets of predictions along label hierarchy (see Appendix D). In Figure 2, increases in set distance show that predictions shift laterally across the label hierarchy, and this shift is affected by classifier expressivity. Therefore, this allows us to contrast classifier predictions along the label hierarchy for classifier selection. By generating and clustering MDS points of base learners of ensembles, we are able to find regions of uncertainty of ensembles that can serve as additional training data (see Appendix F).

## 4 CONCLUSION

GeValDi is able to generate realistic samples of maximal difference, exposing failure modes of classifiers. Further, model expressivity affects the hierarchical set distances of predictions, suggesting that models with higher expressivities are able to learn label hierarchies better. With GeValDi, we are not only able to find failure modes of high-performance classifiers, but also better understand how classifiers have learnt label hierarchies, offering a novel way of evaluating classifiers, especially in the context of ensembling.

## URM Statement

The authors acknowledge that at least one key author of this work meets the URM criteria of ICLR 2023 Tiny Papers Track.

## Acknowledgments

We thank the reviewers for their helpful comments on our work.

KMC gratefully acknowledges support from the Marshall Commission and the Cambridge Trust. UB acknowledges support from DeepMind and the Leverhulme Trust via the Leverhulme Centre for the Future of Intelligence (CFI), and from the Mozilla Foundation. AW acknowledges support from a Turing AI Fellowship under grant EP/V025279/1, The Alan Turing Institute, and the Leverhulme Trust via CFI.

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

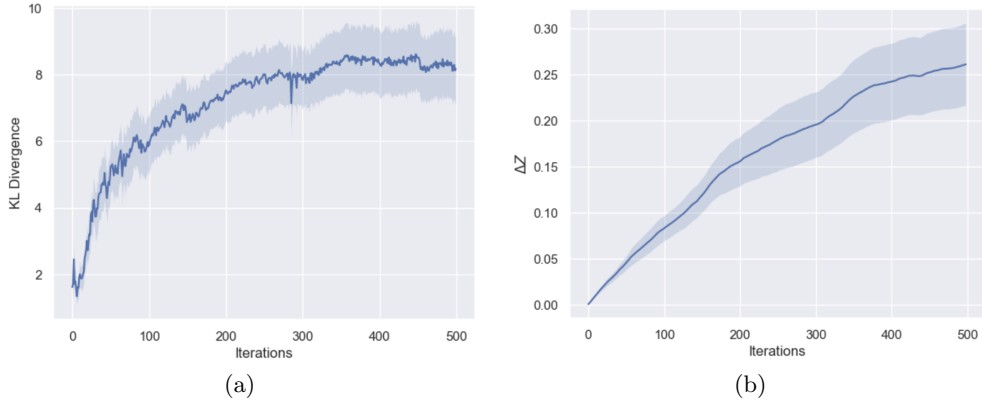

Figure 3: (a) shows the evolution of KL divergence across iterations of MDS algorithm. (b) shows the evolution of $\Delta\mathbf{z}$ across iterations of MDS algorithm

Jianhua Lin. Divergence measures based on the shannon entropy. *IEEE Transactions on Information Theory*, 37(1):145–151, 1991.

Zhuang Liu, Hanzi Mao, Chao-Yuan Wu, Christoph Feichtenhofer, Trevor Darrell, and Saining Xie. A convnet for the 2020s. 2022. doi: 10.48550/ARXIV.2201.03545. URL https://arxiv.org/abs/2201.03545.

Kihyuk Sohn, Honglak Lee, Xinchen Yan, and Min Yu. Learning structured output representation using deep conditional generative models. *arXiv preprint arXiv:1502.08029*, 2015.

Christian Szegedy, Wei Liu, Yangqing Jia, Pierre Sermanet, Scott Reed, Dragomir Anguelov, Dumitru Erhan, Vincent Vanhoucke, and Andrew Rabinovich. Going deeper with convolutions, 2014. URL https://arxiv.org/abs/1409.4842.

Rikiya Yamashita, Mizuho Nishio, Richard Kinh Do, and Kaori Togashi. Convolutional neural networks: An overview and application in radiology. *Insights into Imaging*, 9(4): 611–629, 2018. doi: 10.1007/s13244-018-0639-9.

## A  MDS Algorithm

**Algorithm 1** MDS Algorithm

**Input:** Set of Classifiers $\{p_m(\mathbf{y}|\mathbf{x})\}_{m=1}^M$, Generative Model $p_\theta(\mathbf{x}|\mathbf{z})$ Divergence Metric $D\left[\{p_m(\mathbf{y}|\mathbf{x})\}_{m=1}^M\right]$

**1** Number of Epochs $N$, Learning Rate $\gamma$, Prior Lambda $\lambda_{\text{prior}}$ $\tilde{\mathbf{z}}$ **for** $i \in \{1,...,N\}$ **do**

**2**     $\tilde{\mathbf{x}} \sim p_\theta(\mathbf{x}|\mathbf{z} = \tilde{\mathbf{z}})$

**3**     Classifier Predictions $= \{p_m(\mathbf{y}|\mathbf{x} = \tilde{\mathbf{x}})\}_{m=1}^M$

**4**     $L = D\left[\{p_m(\mathbf{y}|\mathbf{x})\}_{m=1}^M\right] + \lambda_{\text{prior}}p_\theta(\tilde{\mathbf{z}})$

**5**     $\tilde{\mathbf{z}} = \tilde{\mathbf{z}} + \gamma\nabla_{\mathbf{z}}L(\{p_m(\mathbf{y}|\mathbf{x})\}_{m=1}^M, p_\theta(\mathbf{x}|\mathbf{z}))$

**6 end**

**7 return** $\tilde{\mathbf{z}}$;

## B  Validating the MDS Algorithm

In order to make sure the MDS algorithm works as it is supposed to, the path of KL divergence and the evolution of the latent space vector is checked.

Figure 3(a) shows that KL divergence is clearly increasing across iterations consistently across the various experiments. This means that we are actually able to learn samples that our models maximally disagree on, and are not just getting a random draw of samples from the latent space.

Furthermore, we need to understand whether these latent space samples are any different from the initial starting point, or are we just perturbing the initial point to generate 'diverse' samples. We do this by plotting the evolution of the squared difference between the initial starting point and the current latent space point in our MDS algorithm, $\Delta \mathbf{z} = E[\frac{||\mathbf{z}-\mathbf{z}_{\text{init}}||_2}{||\mathbf{z}||_2}]$. As evident in Figure 3(b), we see that the latent space point changes significantly to the initial starting point ($\Delta \mathbf{z} \sim 25\%$) and this shows us that our MDS algorithm actually explores the latent space to find maximally different samples.

## C   Experimental Setup

For our experiments involving the MNIST dataset (Deng, 2012), a Variational Autoencoder (VAE) (Kingma & Welling, 2019) is used as the latent variable model, with Convolutional Neural Networks (CNNs) (Yamashita et al., 2018) as the discriminative classifiers. Note that the CNNs classifiers have classification accuracies of 98.6% and 98.8%.

For experiments involving ImageNet (Deng et al., 2009), pretrained discriminative classifiers and Generative Adversarial Networks (GANs) were used. For the GAN, BigGAN by Deep-Mind (Brock et al., 2018), which generates images from the data distribution that generates the ImageNet dataset, is used. For the classifiers, pairwise comparisons are made between AlexNet (Krizhevsky et al., 2012), GoogleNet (Szegedy et al., 2014), SqueezeNet (Iandola et al., 2016) and ConvNeXt (Liu et al., 2022). Note that the choice of these classifiers provide a wide variety of model capacities that allow us to investigate the effect it has on MDS samples.

To explore the impact of model expressivity, we do 4 experiments (note that M denotes a million parameters): 2 models with similar, but low capacity (SqueezeNet0 - 1.2M, SqueezeNet1 - 1.2M), 2 models with similar, but high capacity (AlexNet - 66.1M, ConvNeXt Base - 88.6M), 2 models with different capacity (SqueezeNet 0 - 1.2M, AlexNet - 66.1M) and lastly, the same 2 models but with the ordering reversed (AlexNet - 66.1M, SqueezeNet 0 - 1.2M). The last experiment explores how changing the ordering of the models in our KL divergence changes the MDS we generate, since KL divergence is non-symmetric.

## D   Prediction Set Distance

We develop a metric to quantify the discrepancy between models' predictions which accounts for *hierarchical structure* in the data space. Specifically, our approach utilizes the pre-defined class hierarchy in ImageNet, known as WordNet (Fellbaum, 1998); this taxonomy defines distance between classes related to their conceptual similarity. Our metric measures the distance between the top $n$ classes predicted by two classifiers, $\mathcal{C}_1$ and $\mathcal{C}_2$. By analyzing the evolution of the distance metric as we progress through the MDS algorithm, we gain insights into how the predictions of two models differ.

$$d(\mathcal{C}_1, \mathcal{C}_2) = \frac{1}{|\mathcal{C}_1||\mathcal{C}_2|} \sum_{c_i \in \mathcal{C}_1} \sum_{c_j \in \mathcal{C}_2} H_w(c_i, c_j) \tag{3}$$

where $|\mathcal{C}|$ gives the size of set $\mathcal{C}$, and $H_w(a, b)$ computes the minimum distance from class $a$ to class $b$ along the hierarchy of classes, which is defined as *half the length of the shortest path from one class to another*.

## E   Additional MDS Examples

In this section, we present several more examples of MDS in Figure 4, with some interesting observations to reinforce the utility of this method.

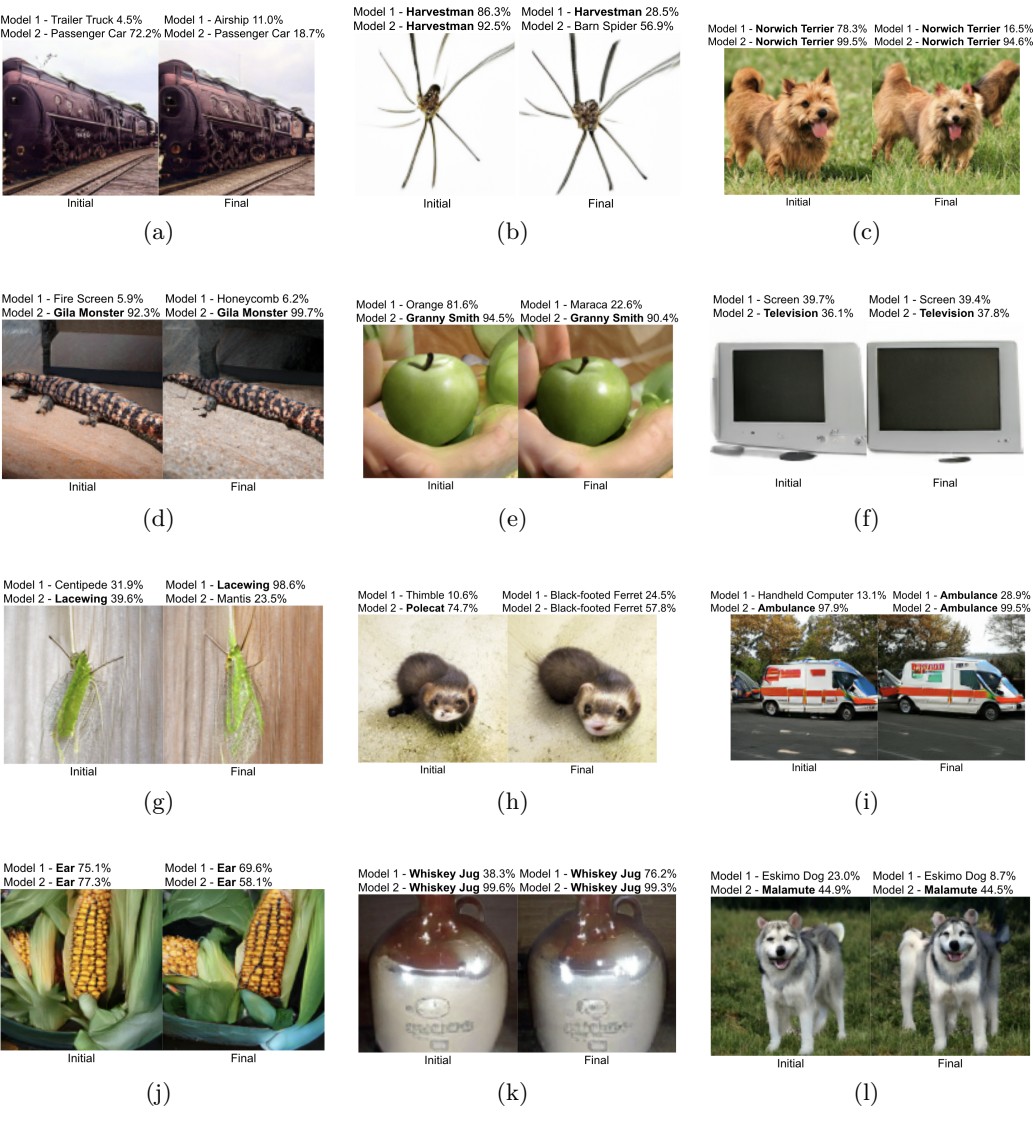

Figure 4: Each figure shows images generated by the GAN before (left) and after (right) the latent space optimisation for Model 1 (GoogleNet) vs Model 2 (AlexNet). Bolded text is used to highlight the BigGAN class-label used to generate samples.

We are able to identify a few types of failure modes of these classifiers, and we look at them in detail here. Firstly, consider Figure 4(b), where both models initially classify the images as "Harvestman" (i.e. the BigGAN class-label). After optimisation, model 2 flips the prediction to "Barn Spider" with more than 50% confidence, which is interesting as it illustrates a failure mode where there are certain input perturbations in the image space that nudge model 2 to flip its prediction. Furthermore, we notice that the set distance between "Harvestman" and "Barn Spider" is only 1.0, which means that flip in class occurred at the same level in the hierarchy.

Contrasting this to example Figure 4(e), where the the predictions evolve from "Orange" and "Granny Smith", to "Maraca" and "Granny Smith", where the set distance increases from 2.0 to 7.0 when model 1 flips its prediction from "Orange" to "Maraca", we see that this failure mode exhibits a large lateral shift of the model predictions along the hierarchy. We note that this category of failure modes are more serious than the previous failure mode because the prediction is more incorrect from the hierarchical perspective in the former case.

As per Figure 4(i), we observe that this latent space optimisation can sometimes point to regions in the image space that improve the model predictions. In other words, in this example, we essentially found perturbations of the image space that nudged both classifiers to correctly classify the image. This behaviour is also observed in Figure 4(k) where the classifiers become more correctly confident in their predictions.

Therefore, these examples point to using MDS to not only find failure modes, but also perturbations in image space that improve model predictions.

## F  MDS ON ENSEMBLES

By applying GeValDi to ensembles, we are able to quantify the diversity of base learners. We know that base learner diversity is critical to the generalisation performance of ensembles (Dietterich, 1970). Training a bagging ensemble (Breiman, 1996) on MNIST, and generating MDS points using a conditional-VAE (Sohn et al., 2015) as the generative model, we are able to quantify how the performance of the ensemble changes between the test dataset and MDS dataset (we know the true label from the conditional class). This performance is depicted in Table 1, where we see a big decrease in accuracy between the test dataset and MDS points. This is why MDS points characterise regions of uncertainty in ensembles.

A natural step is to fully characterise all the uncertain regions of an ensemble by using these MDS points. One way would be to cluster the points and make the clusters represent the regions of uncertainty of the ensemble. For this, we use a Density-based spatial clustering of applications with noise (DBSCAN) algorithm (Ester et al., 1996). The choice of this clustering algorithm comes from the fact that we do not need to choose the number of clusters apriori. This allows us to investigate the number of clusters present in the data in a more general way. However, we still have to tune the radius of search around each point as a hyperparameter, for which we can perform a simple grid search.

For a $10,000$ sample MDS dataset for MNIST, the clusters in the latent space is illustrated in Figure 5, where we plot the clusters on the top two principal component axes for effective visualisation. Comparing the clusters of true labels in latent space and the MDS clusters, we see that MDS clusters form at regions where the label clusters overlap, which agrees with our intuition that these MDS points form regions of uncertainty in our ensemble estimates which naturally arise when the data can be equally from two or more labels.

Examples of MDS points from various clusters is depicted in Figure 6. We note a few interesting aspects here For MNIST, cluster 1, i.e. Figure 6(a), consists of images where the digit can either be a $9, 5, 3$. Similarly, cluster 2, Figure 6(b), consists of digits similar to $1, 7, 2$, and finally, cluster 3, Figure 6(c), consists of digits similar to $8, 3$. By sampling points from the clusters, we can understand specific types of data that the ensemble struggles with, and we can subsequently use these points to further improve the ensemble. We can confirm this by considering the maximum probability classifications of the base model, which are consistent with the possible labels of the images. This suggests that we can use such MDS points as additional training data to further train the ensemble to improve performance.

| | Test Dataset | | | MDS Dataset | | |
|---|---|---|---|---|---|---|
| | Accuracy | AUC | F1 Score | Accuracy | AUC | F1 Score |
| **Bagging** | 93.8 | 0.95 | 0.83 | 65.3 | 0.80 | 0.66 |

Table 1: Comparison of performance of a bagging ensemble in the test dataset and MDS dataset for MNIST.

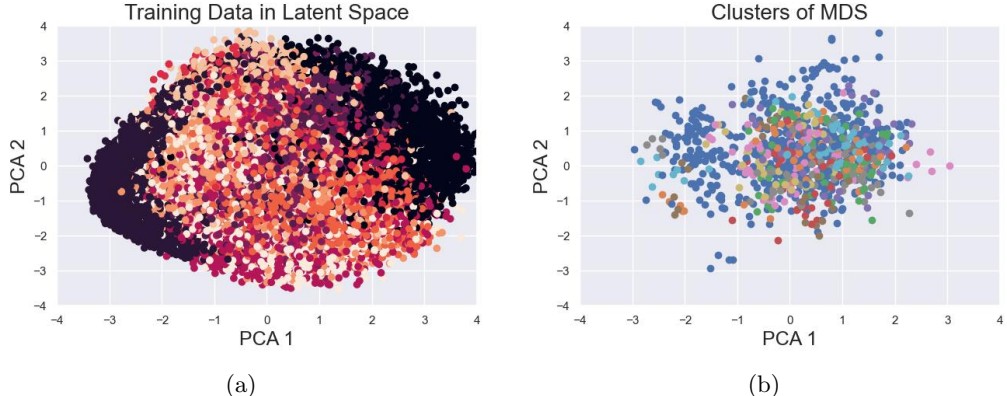

(a)  (b)

Figure 5: (a) MNIST training data clustered by true labels, (b) MNIST MDS data clusters in latent space. Both clusters are plot on the top 2 principal components of MNIST training data.

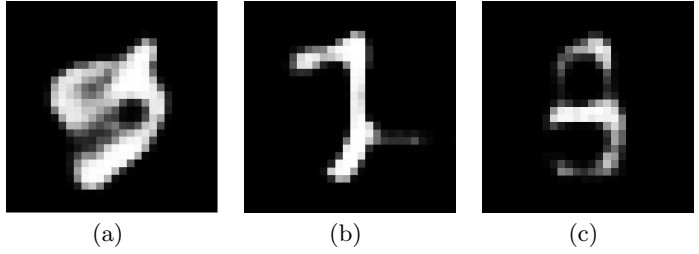

(a)  (b)  (c)

Figure 6: MDS Samples from clusters 1,2,3 respectively for MNIST

