# OpenReview forum: "GeValDi: Generative Validation of Discriminative Models"
_ICLR.cc/2023/TinyPapers — Submitted to Tiny Papers @ ICLR 2023_

### Official Review · Reviewer_KcXu · 2023-03-21

**Confidence:** 4

**Summary Of Contributions:**

The paper presents a generative method, called GeValDi, for validating discriminative classifiers by creating samples where the classifiers maximally differ. The method generates samples in the input space by optimizing in the latent space and is demonstrated to be able to find failure modes of high-performance classifiers.

**Rating:**

Clear, Correct, and Reproducible (CCR): a submission which meets the reviewing criteria

**Strengths And Weaknesses:**

Strengths:
- This paper proposes an interesting generative method that synthesizes data where predictions of two classifiers maximally differ in order to validate discriminative classifiers. The generation search space is constrained by optimizing the latent space.
- The authors demonstrate the failure cases of the classifiers using the generated images before and after the proposed optimization method.
- The paper is overall well-written and easy to follow.

Weakness:
- Evaluation: the authors present some examples of failure cases. However, there lacks an overall metric to quantify the total level of "failure". I saw the authors use KL divergence to measure the distance. But this is more about "comparing" two models pairwisely instead of "validating" the model.
- The use of KLD and lack of actional conclusion: following the above point, the use of KLD can measure the distance of two models, but it is unclear how this distance relates to the classification performance, e.g. given two models having a large difference, which one is better? Also, as the author already pointed out, KLD is asymmetric, so how do we compare which model is better using this metric?

Minor:
Why the authors choose to use AlexNet, GoogleNet, SqueezeNet and ConvNext but not using ResNet, as it is the most commonly used architecture for image classification?

**Suggested Changes:**

1. Would using a symmetric metric, such as the JS-divergence, provide greater clarity than a non-symmetric metric, like KLD, in defining pairwise distance?
2. Distinguishing instead of comparing: the authors should give a more straightforward way to interpret the results in the main text, i.e. which model is better given the calculated distance.
3. The authors could expand their investigation by exploring ways to utilize the generated examples to improve model performance. For example, the authors could consider incorporating misclassified examples into the model's training data to determine if this improves the model's overall performance. This could add more value and interest to the paper.

---

### Official Review · Reviewer_6KRB · 2023-03-27

**Confidence:** 4

**Summary Of Contributions:**

The paper introduces a data-efficient method called GeValDi to validate discriminative classifiers without using validation data. The authors demonstrate how to construct "maximally different samples" and use them to probe the failure modes of classifiers, offering a hierarchically-aware metric to support fine-grained, comparative model evaluation. They provide empirical evidence for the ability of GeValDi to generate "maximally different samples" using ImageNet as a case study.

**Rating:**

Clear, Correct, and Reproducible (CCR): a submission which meets the reviewing criteria

**Strengths And Weaknesses:**

Strengths:
- The paper addresses an important problem in machine learning: how to choose which model to deploy when several models perform equally well on training datasets but differently on unseen data.
- GeValDi offers a data-efficient solution to probing the differences between comparably performing classifiers without using validation data, which is a significant contribution.
- The paper presents a clear and well-organized explanation of the proposed method and its implementation.
- The empirical results demonstrate the effectiveness of GeValDi in generating realistic synthetic samples for which the predictions of two high-performing classifiers differ.

Weaknesses:
- In the early part of the paper, the proposed method was mentioned to find "maximally different samples (MDS)" and evaluate the classifier with MDS. However, there is no discussion in the content on how to distinguish which of the two classifiers is the better model.

**Suggested Changes:**

Suggested changes:

- Can we train a single model that is robust to out-of-distribution by learning a new model using MDS obtained from two models and the original training set? It seems possible to extend this approach to model robustness.

---

### Comment · Area_Chair_yifN · 2023-06-06
**Check for Archival**

This work meets the threshold for archival, contents the URM statement and is deanonymized.

Note to PCs: the default font type seems to be modified.

---

### Meta-Review · Area_Chair_yifN · 2023-04-02

**Recommendation:** Invite to present
**Confidence:** 4

**Metareview:**

This paper studies the efficient evaluation of machine learning models. The authors propose a data-efficient method to probe the differences between comparably performing classifiers without using extra validation data. Specifically, through optimization in the latent space of a generative model, maximally different samples (MDS) in the input space could be constructed to probe the failure modes of classifiers. Experiments on ImageNet are conducted to justify its effectiveness.

All reviewers agree that the paper is well-motivated with good contributions. The reviewers give constructive comments to further improve the paper, such as:
* How to find the better model instead of just comparing;
* Further utilize the generated MDS to improve the model performance.

Overall, based on the review criteria of the ICLR TinyPaper Track, it clearly meets the CCR standard. Please carefully revise the paper following the reviewers' comments, especially the two above-mentioned points. The AC believes it is a good paper after revision.

**Summary:**

This paper studies the efficient evaluation of machine learning models. The authors propose a data-efficient method to probe the differences between comparably performing classifiers without using extra validation data. Specifically, through optimization in the latent space of a generative model, maximally different samples (MDS) in the input space could be constructed to probe the failure modes of classifiers. Experiments on ImageNet are conducted to justify its effectiveness.

**Comments And Feedback To The Authors:**

Comments from AC: what if we have multiple comparable models instead of two models? Is there an efficient way instead of the pairwise comparison?

Please carefully revise the paper (add discussion / extra experiments) following the reviewers' comments.

**Reason For Not Giving A Higher Recommendation:**

I cannot recommend a notable/oral presentation due to remaining concerns.

**Reason For Not Giving A Lower Recommendation:**

Good evaluation by both reviewers.

Well-motivated paper with good contributions. It is a clear accept case.

---

### Decision · Program_Chairs · 2023-04-07

Invite to present